# Assessment of Beeswax Adulteration by Paraffin and Stearic Acid Using ATR-IR Spectroscopy and Multivariate Statistics—An Analytical Method to Detect Fraud

**DOI:** 10.3390/foods13020245

**Published:** 2024-01-12

**Authors:** Konstantinos Chatzipanagis, Jone Omar, Ana Boix Sanfeliu

**Affiliations:** 1European Commission, Joint Research Centre (JRC), 2440 Geel, Belgium; 2Analytical Development, Therapeutics Development & Supply, Discovery Product Development and Supply, Janssen Pharmaceutical Companies of Johnson and Johnson, 2340 Beerse, Belgium

**Keywords:** beeswax, adulteration, attenuated total reflectance, chemometrics

## Abstract

A spectroscopic investigation of beeswax adulteration by paraffin and/or stearic acid was undertaken via Attenuated Total Reflectance Infra-Red spectroscopy (ATR-IR) combined with multivariate statistical analyses. Principal Component Analysis (PCA) was successfully applied for the first time as an exploratory tool for the differentiation among pure beeswax and adulterated beeswax by paraffin and stearic acid with detection limits (LOD) of ~5% and 1%, respectively. Partial Least Square (PLS) modelling was used to build chemometric models based on beeswax/paraffin and beeswax/stearic acid calibration mixtures and subsequently used to predict concentrations of paraffin and stearic acid on a set of unknown test samples. PLS predictions demonstrated that beeswax adulteration by paraffin is much more prominent (74%) than the one by stearic acid (26%) and that commercial beeswax products (candles, pearls, blocks, etc.) are more prone to adulteration (27%) than honeycomb-type samples (12.5%).

## 1. Introduction

Wax made by honeybees (*Apis mellifera* L.) for the fabrication of combs is needed for food storage (honey, and beebread) and brood rearing. Beeswax is a lipid related organic substance generated by the bees by four pairs of wax glands found on the internal part of the 4th to 7th abdominal sternites. More than 300 individual components have been reported in beeswax from various species of honeybee, although their concentrations may slightly vary depending on the honey bee species and their geographical origin [1,2,3,4,5,6,7,8,9,10,11]. Beeswax is generally used in apiculture in the form of honeycombs, candles, cosmetics, pharmaceutics as well as food contact material and/or food additive [12], which means that the adulteration of beeswax by various substances can directly affect the health of animals (bees) and humans.

The most common adulterants are paraffin and stearic acid and may originate from fraud conducted upon beeswax recycling (i.e., deliberate insertion of adulterants to beeswax, known as adulteration) or from random procedures (unintentional use and distribution of adulterated beeswax). In particular, the use of paraffin as adulterant is more frequent than stearic acid, because of its widespread availability, cheap price and relevant physicochemical properties (chemically inert, and colorless). However, both adulterants may cause serious health issues to honeybees that may be in contact with them (from larvae occurring in wax or in industrially produced comb foundations adulterated by cheaper paraffin in order to increase the weight and thus the profit margin) or consume contaminated food (accumulated in beeswax) and to humans through honey and/or honeycomb consumption. In the former case, preliminary investigations carried out by Belgium, France and Germany verified that the existence of stearin/stearic acid is related to health effects on honeybees, referred to as brood development disturbance and rise of larva mortality or impact on bee colonies [10,13,14,15]. On the other hand, humans are considered to ingest the adulterants and their corresponding contaminants transferred from the adulterated honeycomb to the honey or via direct consumption of the adulterated honeycomb. While food-grade stearin is not anticipated to pose any concerns to human health, alkanes found in paraffin can accumulate in various human organs (adipose tissue, spleen, and liver) due to insufficient metabolism and cause damages and/or failures. In addition, mineral oil aromatic hydrocarbon (MOAH) with low alkylation degree present in paraffin can promote tumor [10].

Although there are technical specifications available to assess the quality of beeswax when used as food additive and in pharmaceutics, no regulatory framework exists for its use in apiculture and as a result for human consumption of honeycombs. Nevertheless, purity criteria and relevant specifications to study the authenticity of beeswax used in apiculture have been established by means of spectroscopic and chromatographic analytical methods combined with statistical analyses. 

From a spectroscopy point of view, Fourier transform infrared spectroscopy (FTIR) with an attenuated total reflection (ATR) accessory has been previously employed to detect adulteration in beeswax with approximately or less than 5% of various adulterants such as hydrocarbon waxes, beef tallow and stearic acid [16]. In another study, an analytical method based on ATR data was developed to determine adulteration in beeswax by paraffin, beef tallow, stearic acid and carnauba wax with LOD values less than 3% [17]. In addition, it has been reported that stearic acid, palmitic acid and commercially available stearin, exhibit very similar infrared absorption features and, thus, the spectral ranges characteristic for stearic acid can be also considered to identify palmitic acid and stearin in beeswax [9]. Further progress was achieved by Tanner and Lichtenberg-Kraag who demonstrated that multicomponent adulteration with up to five types of adulterants (paraffin, stearic acid, tallow, carnauba wax and candelilla wax) could be detected as accurately as single component adulteration [18]. Apart from investigations on beeswax samples received from random sources, systematic studies have been also performed to identify percentages of paraffin and stearic acid adulteration on a national level (Belgium), showing that commercial beeswax was more prone to adulteration compared to beekeepers beeswax samples [19]. Based on the above mentioned reports, it is possible to detect beeswax adulteration of less than 3% of these adulterants and their combinations by FTIR-ATR spectroscopy.

In the present study, ATR-IR spectroscopy followed by advanced statistical techniques was applied to determine the adulteration of beeswax by paraffin and stearic acid. A large set of samples containing both commercial beeswax products (candles, pearls, blocks, etc.) and honeycombs was analyzed, with the latter being used to establish baseline levels for the contents of paraffin and/or stearic acid to determine the extent and practices of beeswax adulteration. To the best of our knowledge, this is the first study where a fully comprehensive statistical evaluation is performed by combining PCA and PLS analyses for the exploration of beeswax adulteration and estimation of concentrations of adulterants, respectively. It is important to highlight here that although the use of PLS has been already discussed in the literature, there is a lack of any existing report that employs PCA on infrared spectroscopic data for the investigation of beeswax adulteration. It is intended through the present work to bridge this gap in knowledge and demonstrate that PCA is a fast and highly robust classification tool, which, upon combination with infrared spectroscopy, can serve as a reliable screening method to determine the presence of paraffin and stearic acid in beeswax. 

## 2. Materials and Methods

### 2.1. Test Set 

A total of 110 beeswax samples were purchased and subjected to investigation for adulteration by paraffin and stearic acid. From the total number of samples, 74 specimens were bought online, 22 from commercial sources and 14 from professional beekeepers. They were purchased from various countries around Europe, such as Germany, Austria, Spain, Greece, Belgium, France, Netherlands, Italy, Bulgaria, Turkey and Hungary. In terms of sample coloring, the vast majority of the specimens were found to be yellow (light and dark), some of them were white and a few of them were seen as brown/black. The beeswax samples investigated in the present study are broadly divided in two categories: (i) 78 beeswax products in the form of candles, pearls, blocks, sheets, etc., and (ii) 32 honeycombs containing honey. For the latter category, honey was extracted using hot water and the samples were separated into beeswax from the foundation (middle part) and beeswax from the sidewalls. These sidewalls can be used to establish baseline levels for the contents of paraffin and stearic acid, as it may be assumed that they were built by the bees and were not subjected to any other manipulation and thus contained only typical background levels of the adulterants under investigation. As a result, two subsamples were formed for every honeycomb sample, one containing the sidewalls and the other including the foundation part (labelled as XXMP, XX: sample number). 

### 2.2. Calibration Set

Two reference paraffin samples and two reference stearic acid samples were purchased. For both reference types, one sample was purchased online and the other from Kahlwax (NL) company, a leading specialist in the production of natural wax and several other products. The two paraffin samples were checked by IR spectroscopy and their corresponding spectra were almost identical. The same observation was also made for the stearic acid samples.

The calibration set was prepared by mixing three different non-adulterated beeswax samples and subsequently spiking them with paraffin and stearic acid in increasing amounts. Τhe absence of adulteration in the three beeswax samples was verified by PCA, where it was observed that these samples were projected in a big cluster away from the adulterants and together with honeycomb sidewalls that are considered pure, since the latter would only contain background levels of the two adulterants. For paraffin calibration samples, beeswax was spiked with paraffin at: 0, 5, 10, 15, 20, 25, 50 and 100% (*w*/*w*) denoted as CP1, CP2, CP3, CP4, CP5, CP6, CP7, CP8 and for stearic acid calibration samples, beeswax was spiked with stearic acid at: 0, 0.5, 1, 5, 15, 25, 50 and 100% (*w*/*w*) denoted as CA1, CA2, CA3, CA4, CA5, CA6, CA7, CA8. For each calibration sample, the solid mixture was homogenized by melting at 85 °C for 1 h and re-solidified by cooling the mixture at room temperature.

Note that different concentrations were considered for the two calibration sets, particularly for spiking levels below 15%. This was primarily carried out to account for the different LOD values reported for stearic acid (1%) and paraffin (5%), respectively. For this reason, spiking with stearic acid started at very low concentrations (<1%), whereas in the case of paraffin, spiking at such low levels would not allow differentiation from pure beeswax. Τhus, the first paraffin spiking level was established at ~5%. Above 15%, approximately the same concentrations were considered for both adulterants, covering the entire range needed for calibrations. The calibration samples and their nominal concentrations are shown in detail in Appendix A.

### 2.3. Infrared Spectroscopy

An Alpha II compact FTIR spectrometer (Bruker, Ettlingen, Germany) coupled with a platinum ATR module containing diamond crystal as the ATR element located on a heating stage and a temperature controlled DLaTGS-detector was used to conduct infra-red (IR) measurements in the range of 400–4000 cm^−1^ using 24 scans at 4 cm^−1^ spectral resolution. In order to perform spectral acquisition in the liquid state, the samples were heated at 85 °C on a hotplate to achieve complete melting and a drop of the sample melt was uniformly applied on the surface of the diamond crystal, which was also heated at the same temperature to prevent specimen solidification during measurement. A background spectrum was recorded prior to each sample measurement and a blank spectrum was subsequently taken after cleaning the crystal to ensure that the cleaning process was successful before the next measurement. Spectra were recorded in duplicates for each sample.

### 2.4. Statistical Analysis

Multivariate statistical analysis was performed using the software package SIMCA^®^ 17 (Umetrics, Malmö, Sweden), which is named after the widely recognized classification technique known as Soft Independent Modelling of Class Analogy. SIMCA^®^ 17 is a well-known and user friendly software tool designed to perform multivariate data analysis in process analytical technologies (PAT). This version is complemented by an upgraded spectroscopic functionality through the integration of spectral analysis, process modelling and enhanced data pre-processing procedures in one effective solution. Furthermore, the introduction of a new calibration wizard feature, where it is possible to directly split the samples into calibration and validation groups, has greatly facilitated the development of robust calibration and prediction models. Considering these aspects and the fact that the present study is based on infrared spectroscopy, SIMCA^®^ 17 software was chosen to conduct both untargeted and targeted analysis on the adulteration of beeswax.

The first step of the statistical evaluation involved PCA for exploratory spectral analysis of the entire test set. This step constitutes a classification analysis to investigate non-adulterated and adulterated samples in an untargeted manner. In the second step, targeted analysis was performed by applying PLS to develop calibration models using the calibration samples and these models were subsequently used to predict the concentration levels of paraffin and stearic acid in the test set. Prior to any statistical analysis, spectral treatment was performed by mean-centering and standard normal variate (SNV) across the entire wavenumber range.

## 3. Results and Discussion

### 3.1. Infrared Spectra of Beeswax and Common Adulterants

Figure 1 shows the infrared spectra of beeswax and its two most common adulterants, paraffin and stearic acid. It is observed that all three chemical substances exhibit IR absorption in the spectral ranges around 400–1800 cm^−1^ and 2800–3000 cm^−1^.

On one hand, significant variations in the number, frequency and relative intensities of the IR bands are observed among the different substances in the 400–1800 cm^−1^ region. On the other hand, all substances share a very common strong doublet band in the high frequency region at 2800–3000 cm^−1^ that is assigned to CH_2_ vibrations with similar relative intensities [20,21,22,23,24,25]. As a result, the pure beeswax spectrum is modified in accordance to the presence of specific adulterants and it is mainly in the 400–1800 cm^−1^ frequency range that these modifications are observed.

However, careful visual inspection of the untreated IR spectra of the test samples revealed that samples 16 and 22 exhibit some differences in the 1000–1780 cm^−1^ range compared to the rest of the test set, indicating that these two samples may contain additional adulterant(s) to paraffin and/or stearic acid. Figure 2 shows the untreated IR spectra of all test samples together with the reference spectra of tri-stearin (Tr) and beef fat (BF), the latter substances being frequently used in beeswax adulteration. Tr was purchased as an industrial product, while BF was bought from a local butcher shop and they were both measured under the same conditions described in Section 2.3.

The IR spectra of tri-stearin and beef fat appear very similar and in agreement with the corresponding spectra reported on beef tallow [17,18]. To facilitate the comparison, the spectra of samples 16, 22, Tr and BF are highlighted in various colours, whereas the spectra of the rest of the test samples are represented in grey (also seen as black due to spectral overlap). The 1700–1780 cm^−1^ range is illustrated separately in the graph below for better visualization.

Three spectral features located at 1155, 1233 and 1746 cm^−1^ are observed in samples 16, 22, Tr and BF, whereas similar features in the other test specimens are found at 1171, 1243 and 1738 cm^−1^. In addition, samples 16 and 22 demonstrate a shoulder at 1099 that is also present in Tr and BF, while this feature is absent in the spectra of the other test samples. Therefore, it seems that these two samples exhibit spectral similarities with the two reference substances and thus they were also included in the PCA.

### 3.2. Principal Component Analysis (PCA)

Following the pre-processing of the IR spectra, a PCA was carried out with three principal components that can explain ~96% of the total variance. Figure 3 shows the scores plot of the first two principal components (t2 vs. t1) for the entire set of test and reference samples, explaining ~88% of the sample variance. A ‘’leave-out’’ internal cross-validation procedure was employed during model fitting and no further external validation was performed.

The two reference samples of stearic acid are located (black scores) at the top of t2, whereas at the very left of t1 there are the two reference paraffin samples (red scores) and a number of other beeswax samples that could be evidently seen as largely adulterated with paraffin. The big cluster observed in the middle of t1 contains beeswax samples that may be considered as ‘non-adulterated’ due to their distance to the studied adulterants, while reference Tr and BF samples (blue scores) are situated at the right part of t1. It is also observed that honeycomb samples (sidewalls and middle parts) are largely projected within the big central cluster, indicating that they contain either no or very minor concentrations of adulterants, as expected particularly for the sidewalls.

Another interesting observation is that samples 17, 26 and 62 do not belong to any of the four clusters, denoting partial adulteration by one or more adulterants. In particular, sample 26 is located between the clusters of pure beeswax and stearic acid, which confirms partial adulteration with stearic acid. Under the same principle, partial adulteration by paraffin is observed for sample 62, whereas sample 17 seems to contain a mixture of paraffin and stearic acid.

Furthermore, evaluation of the third principal component against the first two components demonstrates an additional ~9% contribution in the explained variance, as shown in Figure 4a,b. Note that in this graph, all references and test samples are shown in green apart from samples 16 and 22 that are highlighted in red for further discussion. To correlate with the clustering seen in Figure 3, non-adulterated (pure) beeswax is shown inside the purple circle and stearic acid inside the blue circle, whereas reference paraffin and highly paraffin adulterated beeswax samples are illustrated within the red circle.

According to these PCA plots, samples 16 and 22 (scores highlighted in red) appear somehow different from the pure beeswax samples due to their separation from the tight cluster, which is in line with the findings of Figure 2. In particular, the two samples are projected between the pure beeswax cluster and the two reference samples of Tr and BF, denoting the presence of the latter substances. However, samples 16 and 22 do not align perfectly between pure beeswax and Tr/BF, but they appear shifted to the left (Figure 4a), which indicates that they may also contain moderate amounts of paraffin pulling them to the left. Similarly, a small shift to the right may be observed in Figure 4b that would imply minor presence of some stearic acid as well.

Although the spectra of the two reference samples are very similar, the width of the band at 720 cm^−1^ appears larger in beef fat than the one seen in tri-stearin and the test samples, which indicates that samples 16 and 22 are more likely to contain Tr. Of course, this does not exclude the presence of BF in these two samples.

The potential presence of Tr together with paraffin and stearic acid was independently tested by performing a comparison between measured and calculated spectra computed from reference samples (beeswax, paraffin, Tr, and stearic acid) with appropriate weighting factors. An example of such calculation is shown in Appendix A, indicating the presence of paraffin, stearin and stearic acid.

To obtain a better insight into the adulteration of beeswax with paraffin and/or stearic acid, the PCA was performed again by including the prepared calibration samples, as demonstrated in Figure 5a. It can be observed that a systematic increase in either paraffin or stearic acid content results in an equivalent spread of the relevant calibration samples away from the tight clustering (pure beeswax) towards pure paraffin and stearic acid reference samples.

Figure 5b shows an enlarged view of the tight clustering located at the centre of Figure 5a for better visualization. Calibration samples CP1, CA1, CA2 and CA3 are located within the centre of the cluster, whereas calibration sample CP2 is found to be at the left side of the cluster.

This observation shows that beeswax samples containing up to 5% paraffin (CP2) and 1.14% stearic acid (CA3) are overlapping with pure beeswax and no differentiation can be made below these contents using infrared spectroscopy. Hence, these concentration values define the LOD values of paraffin and stearic acid in this study and as such, they can be considered threshold values for the determination of beeswax adulteration.

Based on the above, PCA not only provides a qualitative confirmation on the adulterated beeswax samples, but it can also provide quantitative insight on the respective adulteration levels. This is the first study demonstrating that the combination of IR spectroscopy with PCA can serve as a powerful and fast tool for the investigation of beeswax adulteration.

### 3.3. Partial Least Squares (PLS)

Following the investigation of beeswax adulteration by PCA, the levels of paraffin and stearic acid concentrations were estimated by PLS. Statistical models were developed for both adulterants using the calibration samples and validation was performed via ‘’leave-out’’ internal cross-validation approach, which is generally applied when the available number of samples is not large (<40) [26]. The calibration curves for both paraffin and stearic acid are well described by a linear fitting with R^2^ > 0.99, indicating the high quality of the two calibration models. Hence, no external validation was considered to further assess the robustness of the produced models. These curves were also used to determine the values of LOD for both adulterants via the following equation:(1)$$\mathrm{LOD}=\frac{3.3\times SD}{s}$$
where *SD* is the standard deviation of the fitting residuals and s is the slope of the regression line. Based on Equation (1), the calculated LOD values were 5% and 1.1% for paraffin and stearic acid, respectively. These values are in agreement with previously reported LOD values [16,18].

The calibration models were subsequently used to predict the concentrations of paraffin and/or stearic acid in the test specimens, as shown in Table 1. Honeycomb-type samples are represented in bold/italic. These PLS predictions are in general correlation to the PCA observations seen in Figure 3 and Figure 5, where it is possible to obtain a quantitative idea of the concentrations of the two adulterants in the test samples, since references and/or calibration samples are also included in the PCA. This is an additional indication that PLS predictions are valid and thus so are the corresponding calibration models used to perform these predictions.

On one hand, it is observed that several samples seem to contain very high amounts of paraffin (>80%), while two of them are found to contain ~50%. However, most of the investigated samples exhibit either low amounts of paraffin (<10%) or no paraffin. It is also interesting to note that those specimens that are characterized as highly adulterated (>80%) are beeswax products in the form of pearls and blocks, whereas honeycombs were generally perceived as pure. Few exceptions were solely observed for some honeycomb middle parts (comb foundations) with a minor paraffin content (<10%).

On the other hand, samples 17 and 26 exhibit moderate to high amount of stearic acid (10–40%), while 66 and 71 show minor concentrations of stearic acid (<5%). These samples are beeswax products in the form of candles and blocks. Three honeycomb specimens (87, 87MP and 98MP) were found to have traces of stearic acid very close to the LOD (<3%), while no traces are seen for the rest of the specimens.

PLS prediction ranges for adulterated beeswax with paraffin (>5%) and stearic acid (>1%) are presented in Table 1. According to Table 1, paraffin is responsible for ~74% of the total adulteration observed in the test samples, while the remaining 26% is assigned to stearic acid. Hence, paraffin appears to be a more frequent adulterant in beeswax compared to stearic acid, which is in agreement with previous work conducted by Serra Bonvehı’ and Orantes Bermejo [11]. Another investigation comprising beeswax samples from 15 different European countries has shown that the presence of stearin/stearic acid was predominant only in Belgium and Netherlands [27]. Our findings are also somehow in line with the outcome of this investigation, since samples from various different countries were considered in our study with only a small number of them originating from either Belgium and/or Netherlands and thus it is not surprising that paraffin dominates over stearic acid as the main adulterant. Nevertheless, even for the samples originating from either Belgium and/or Netherlands, no particular trend of increased adulteration by stearic acid compared to paraffin was observed.

Moreover, 12.5% of the honeycomb type samples were adulterated by paraffin and/or stearic acid, whereas beeswax products were found to exhibit approximately 27% of adulteration. It is thus observed that beeswax products tend to be more prone to adulteration compared to honeycombs, the latter being purchased either online or directly from beekeepers. This is in accordance with the investigation performed by El Agrebi et al., where it was demonstrated that adulteration in commercial beeswax was approximately 3.5 times larger than in samples coming directly from beekeepers [19]. Moreover, for the honeycombs found adulterated in this study, it is found that both paraffin and stearic acid contents are generally low and it is mainly the middle parts that exhibit some adulteration, whereas the sidewalls appear pure.

## 4. Conclusions

The present work demonstrates the combination of ATR-IR spectroscopy with both PCA and PLS multivariate statistics for the investigation of beeswax adulteration. PCA was successfully employed for the first time to identify clusters of authentic and suspicious beeswax samples and to obtain some insight on the corresponding adulteration levels, upon inclusion of calibration samples in the analysis. Beeswax authenticity was mainly based on the assessment honeycomb samples that were separated into sidewalls and middle parts for further studies. Honeycomb samples were largely located within a big cluster that is well displaced from reference samples (paraffin, stearic acid, Tr, and BF), denoting that this area is characteristic of pure beeswax. Similarly, any commercial beeswax product (pearls, blocks, etc.) projected inside this cluster is also considered to be non-adulterated.

In addition, the presence of few paraffin and stearic acid calibration samples inside this cluster indicates LOD values of ~5% for paraffin and ~1% for stearic acid in native beeswax. PCA also revealed additional adulteration by stearin for samples 16 and 22, when an additional (third) principal component was taken into account.

It is important to point out that this is the first study reporting on the application of PCA for the determination of beeswax adulteration as well as the investigation of honeycomb samples into sidewalls and middle parts individually.

On the other hand, PLS predictions provided estimations of the concentrations of paraffin and/or stearic acid found in beeswax, confirming that beeswax products are regularly more prone to adulteration (27%) than honeycombs (12.5%) as already seen by PCA and that paraffin presence is three times more prominent than the presence of stearic acid. These findings are in agreement with previous works, as described in Section 3.3.

Overall, the combination of ATR-IR spectroscopy with PCA and PLS appears to be a reliable and fast method to study the adulteration of beeswax by paraffin and stearic acid.

## Figures and Tables

**Figure 1 foods-13-00245-f001:**
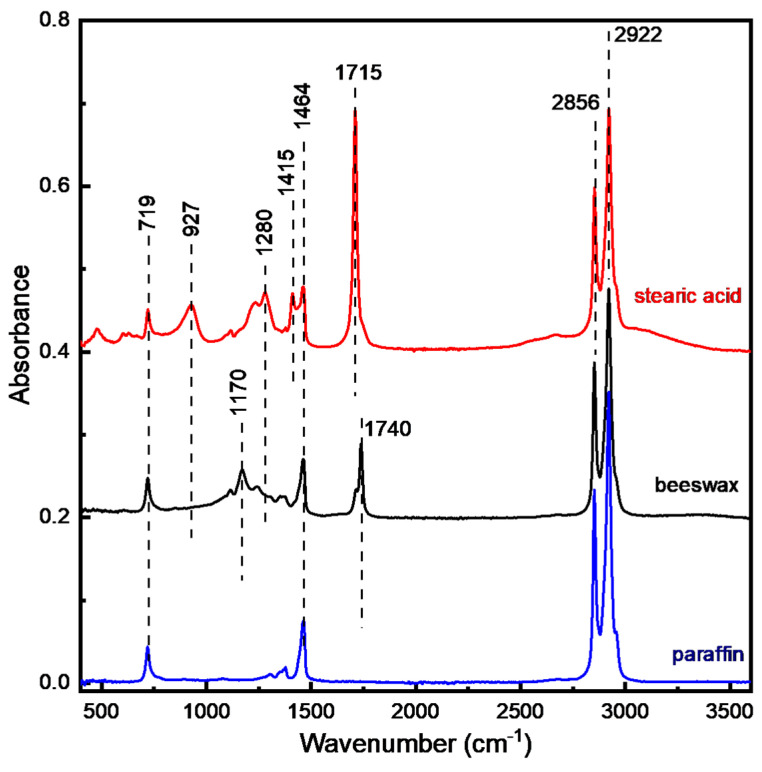
IR spectra of paraffin, beeswax, stearic acid. Spectra (except paraffin) are shifted for clarity.

**Figure 2 foods-13-00245-f002:**
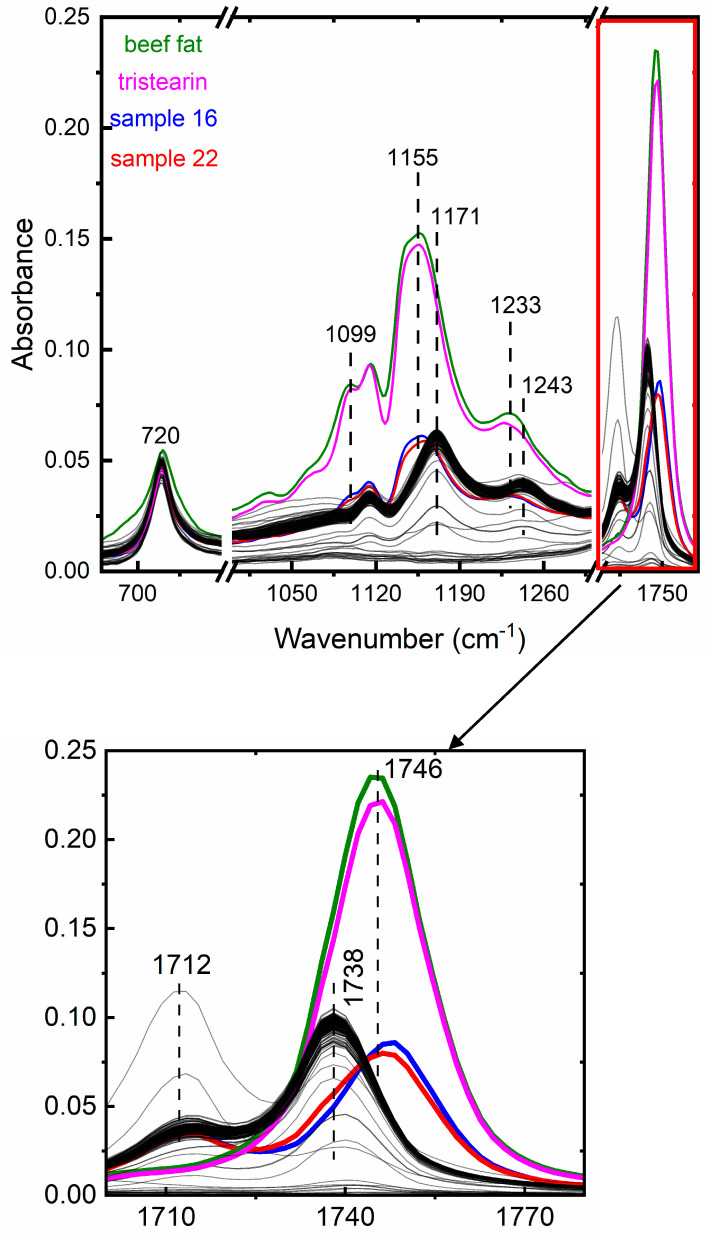
IR spectra of beeswax samples, BF and Tr. Spectra of BF, Tr, samples 16 and 22 are highlighted in various colours. All other test samples are depicted in grey (also seen as black due to spectral overlap). Zoom of the area above 1700 cm^−1^ is shown in the graph below.

**Figure 3 foods-13-00245-f003:**
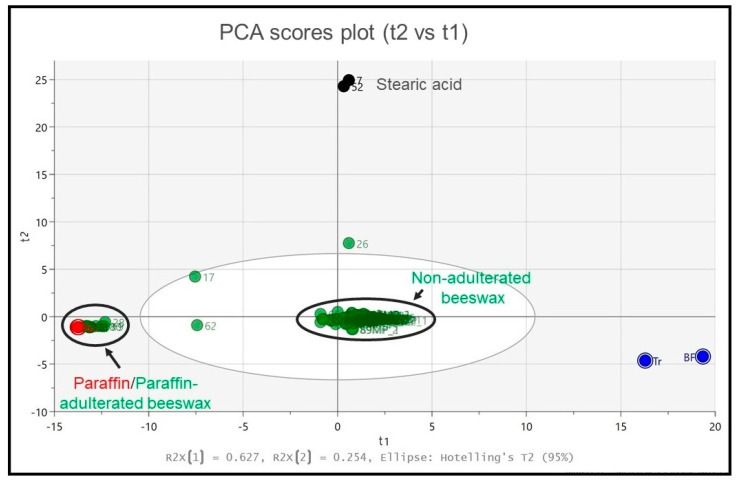
PCA scores plot (t2 vs t1). Reference stearic acid (samples 7 and 52) are shown in black, reference paraffin in red, beeswax samples are illustrated in green and Τr/BF references are shown in blue. Left black circle: Highly paraffin adulterated beeswax samples (in green) are projected together with reference paraffin (in red). Central black circle: Νon-adulterated (pure) beeswax samples.

**Figure 4 foods-13-00245-f004:**
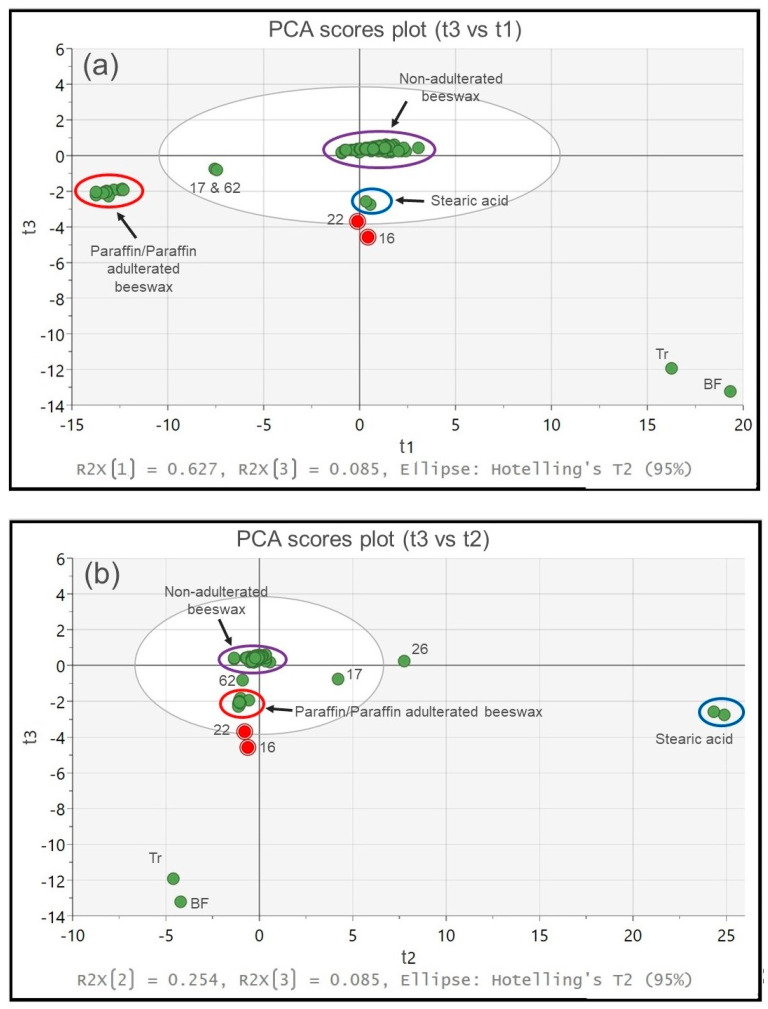
(**a**). PCA scores plot (t3 vs. t1). (**b**). PCA scores plot (t3 vs. t2). Purple circle: Non-adulterated (pure) beeswax samples. Blue circle: Stearic acid samples. Red circle: Reference paraffin and highly paraffin adulterated beeswax samples. Scores corresponding to samples 16 and 22 are highlighted in red for better visualisation.

**Figure 5 foods-13-00245-f005:**
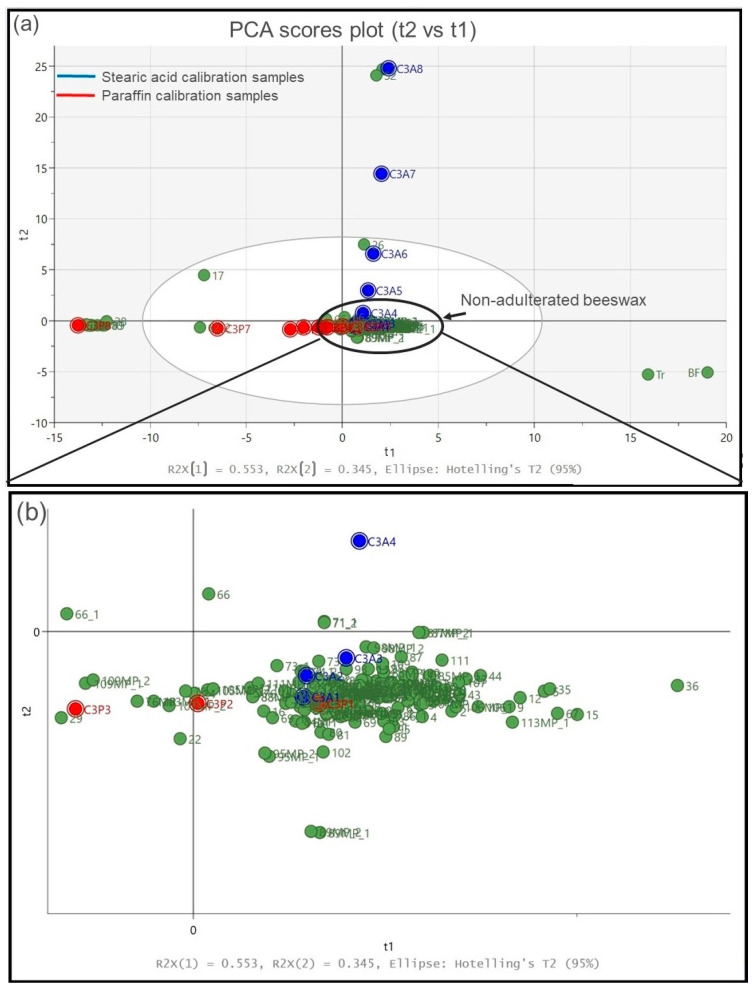
(**a**). PCA scores plot (t2 vs t1) of beeswax, reference and calibration samples. Stearic acid calibration samples are shown in blue, paraffin calibration samples in red and beeswax, Tr and BF in green. (**b**). Enlarged view of the tight clustering area circled in (**a**).

**Table 1 foods-13-00245-t001:** PLS predictions for paraffin and stearic acid concentrations.

Paraffin	5–10%	10–15%	15–75%	>75%
	24, 66, ***76***, ***76MP***, ***103MP***, ***105MP***	29, ***109MP***	17, 62	8, 10, 11, 13, 18, 19, 28, 31, 32, 33, 34, 51
**Stearic acid**	**1–5%**	**5–20%**	**20–75%**	**>75%**
	66, 71, ***87***, ***87MP***, ***98MP***	17	26	**-**

## Data Availability

Data are contained within the article.

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
