# Peer review of "Assessment of Beeswax Adulteration by Paraffin and Stearic Acid Using ATR-IR Spectroscopy and Multivariate Statistics—An Analytical Method to Detect Fraud"

_foods, 2024, doi:10.3390/foods13020245_

Round 1

Reviewer 1 Report

Comments and Suggestions for Authors

In this study, FTIR-ATR spectroscopy, coupled with multivariate data analysis techniques, was employed to identify the presence of paraffin and stearic acid as adulterants in commercial beeswax products. While the study is intriguing, certain elements are missing in the manuscript that need to be addressed. Please refer to my suggestions below.

•          The materials and methods section is not very descriptive. It should be given in more detail.

•          Line 88: “A high number of beeswax samples.” How many samples are there in total? How many did you purchase from online stores? How many from commercial stores and beekeepers? Did you purchase them from which countries? Be much more specific. All the answers should be added to the manuscript.

•          Lines 102-103: “Reference standards of paraffin and stearic acid were purchased online and from a laboratory in the Netherlands.” How many reference standards of paraffin and stearic acid did you buy? Purchased from a laboratory? All the answers should be added to the manuscript.

•          Lines 103-104: “Three different non-adulterated beeswax samples” How do you know they are non-adulterated? Where did you get them? All the answers should be added to the manuscript.

•          Lines 105-107: You spiked the samples in the concentration levels of 0, 5, 10, 15, 20, 25, 50, and 100% (w/w) for paraffin and 0, 0.5, 1, 5, 15, 25, 50 and 100% for stearic acid. Why did you choose different concentrations of stearic acid and paraffin? It should be added to the manuscript.

•          Lines 126-127: What is SIMCA 17? It should be given in full name and then in abbreviation since you have never talked about SIMCA before. Also, a description of SIMCA and why you choose to use it in your manuscript needs to be explained.

•          The researchers have applied a leave-one-out approach to validate the PLSR analysis, but did they do external validation? Also, is there any external validation for the classification?

•          Another essential point was that there was no literature comparison in the discussion section. Are this study's findings (performance) comparable with the literature?

Comments on the Quality of English Language

The manuscript exhibited some minor problems with the English language. Therefore, it needs to be re-read and corrected by the authors.

Reviewer 2 Report

Comments and Suggestions for Authors

Round 2

Reviewer 1 Report

Comments and Suggestions for Authors

The authors have made substantial revisions to the manuscript based on my feedback. However, they mentioned they couldn't find any information regarding the "full name" of SIMCA, a widely recognized supervised classification technique. I recommend that they conduct a more through search to acquire relevant information and incorporate it into the manuscript.

Additionally, neglecting external-validation for the calibration models poses a concern.

Comments on the Quality of English Language

The English language is fine, just minor points.
